# Gamble for the needy! Does identifiability enhances donation?

**Marc Wyszynski** [1]*, **Adele Diederich**[2], **Ilana Ritov**[3]

**1** Department of Psychology & Methods, Jacobs University, Bremen, Germany, **2** Department of Life Sciences & Chemistry, Jacobs University, Bremen, Germany, **3** Department of Psychology, The Hebrew University of Jerusalem, Jerusalem, Israel

\* m.wyszynski@jacobs-university.de

## Abstract

To investigate how neediness and identifiability of a recipient influence the willingness of a donor to invest resources in charity-like lotteries we propose a new game, called 'need game'. Similar to the dictator game, the need game includes two players, one active player (the donor or dictator) and one passive player (the recipient). Both players require a minimum need ($N_D$ and $N_R$), expressed in terms of points. The donor is endowed with $K_D$ points and must retain at least $N_D$ points, i.e., the need, with $N_D < K_D$, at the end of the game with $n$ rounds. The recipient starts with $K_R$ points and must end the game with at least $N_R$ points, i.e., the need, with $K_R < N_R < K_D$. The donor is asked to choose one of three different amounts from $K_D$ to place a bet on a lottery. If won, the gain is added to the endowment. If lost, the recipient receives the points. The recipient is paid only when his/her need threshold is obtained; likewise the donor gets paid only when his/her need threshold is maintained. The main focus here is on need of both players ($N_D = N_R = 2, 200$, and $ND = N_R = 0$ serving as baseline control) and the identifiability of the recipient (no information, described by text and picture, and physical presence). We probe whether the amount invested by the donor depends on need and identifiability of the recipient. In addition, we include the framing of the game as gain or loss, different probabilities to win/lose, and different time limits as covariates. We found that each of these factors can play a role when investing in charity-like lotteries.

## Introduction

In modern welfare states, wealth is typically redistributed from the better off to those with less income. This may occur through various means such as progressive income tax levels or tax-based social benefits. The proportions and thereby the amounts transferred within a society are determined by legislation within a given period. Another means often used by welfare states for redistribution of resources is running charity lotteries. Charity lotteries such as the Deutsche Fernsehlotterie and Aktion Mensch in Germany, Mifal HaPais in Israel, or The National Lottery Community Fund in the UK raise money by selling lottery tickets to the public to fund their social programs and support people in need. To incentivize donors, a small

**Data Availability Statement:** Materials, instructions for participants, data, and the R code used for the analysis are publicly archived on osf. io: https://doi.org/10.17605/OSF.IO/QDMUJ.

**Funding:** - Initials of the authors who received each award: AD. Grant numbers awarded to each

author: Grant DFG FOR2104 (Need-based justice and distributive procedures), DI 506/13-1,2 - The full name of each funder: Deutsche Forschungsgemeinschaft. - URL of each funder website: https://gepris.dfg.de/gepris/projekt/259014267 - The funders had no role in study design, data collection and analysis, decision to publish, or preparation of the manuscript.

**Competing interests:** The authors have declared that no competing interests exist.

proportion of the lottery revenue is distributed to them in form of monetary or other prizes. Furthermore, to invite more people to participate in these lotteries, they are often advertised by showing individuals or groups who benefit from specific programs.

Even when tax-payers tend to complain about giving up too much of their own money for the welfare system, they often are generous when it comes to freely donating money to people in need. Donations may come in different ways, for instance, by sending money directly to a charity or by joining a state or charity lottery. For the former, there is no monetary return for the donor, whereas, for the latter, there is a chance to win a prize. But do people who buy charity lottery tickets think of their purchase as making a donation, or do they simply see it as an opportunity to gamble in order to potentially increase their own wealth? To the extent that charity lottery is perceived differently from a plain lottery, peoples' preferences among gambles may depend on factors that typically affect charitable giving.

The main goal of the present study is to investigate various factors that may influence the amount of resources people are willing to invest in charity lottery-like gambles. To this end, we propose a new game, called 'need game', which is a modification of the dictator game [1] and includes a lottery (Note, that a real charity lottery obviously differs in many aspects from an experimental lottery in the lab as outlined later, and we use the example mainly for motivational reasons. However, the experimental paradigm shares some similarities with charity lotteries: 1) the involvement of needy recipients; 2) donors join a rsiky gamble; 3) donors can choose how much they are willing to invest; 4) recipients may receive a portion of the money invested by donors, and 5) recipients take a passive role, i.e. they are not able to influence the outcome of the lottery). In particular, we are interested in whether 1) a defined need for both the donor and the recipient and 2) the description of the recipient in need affects the amount invested. To compare the study to previous research, we also investigate the influence of 3) the probability of winning the lottery; 4) the framing of the lottery; and 5) different time limits for making a choice on choice behavior. We start by describing some of these concepts and then state our specific hypotheses.

Need has been defined as an objective lack of resources that are necessary to maintain physiological and mental health, as an actual or felt state of deprivation, or as the incapability to partake in the commonly accepted activities of the community (for an interdisciplinary perspective on needs and need-based justice see the contributions in the edited volume by Kittel & Traub [2]). Irrespective of its theoretical conception, need seems to serve as an important reference point for allocating resources. For instance, in a questionnaire experiment, Schwinger & Lamm [3] and Lamm & Schwinger [4] showed that participants allocated a significantly higher share of money to more needy persons even if they were responsible for their higher need. Here, we assume that both, the donor and the recipient have needs—expressed in terms of a minimum amount of points—that is required for living. At the beginning of the experiments, the donor has some endowment worth more than the minimum need. The recipient has no endowment but her/his minimum need can be reached through the donor's repeated actions.

The standard dictator game consists of two players: an active player (the dictator), who determines how to split an endowment between her/himself and a second, passive player, who has no influence on the outcome of the game. Numerous dictator game experiments have been conducted. Overall, generosity seems to be large, i.e. more than 50% of the dictators are willing to share their endowment (see e.g. [5]). Conducting a meta-analysis with 131 studies and 616 treatments, Engel [6] found that, for instance, deserving recipients get more and identified dictators are more generous. For our purpose, we modify the standard dictator game in several ways. First, we define a minimum need level for both, dictators, here we keep calling the person *donor*, and recipients. The minimum need level of the recipient can be reached only by

repeated actions of the donor, and therefore, the game must run for several rounds with the same participant. Second, the game is embedded into a lottery to produce risky-choice. That is, the donor decides on how many points to invest in (share with) the recipient (similar to how many lottery tickets are bought), however, the recipients will get them only with a prede-fined probability (similar to the donor's lost bet). We will describe the game, which is called 'need game', in detail in the method section.

To the extent that people think of buying charity lottery tickets as donation, the characteristics of the recipient should affect their decisions. In particular, the identifiability of the recipient is expected to make a difference. In his economic analysis on the value of life, Schelling [7] distinguished from the beginning between identified and statistical lives. His hypothesis that people react differently to identified and statistical lives has been supported by several experiments (e.g. [8–12]). The effect is nowadays called "Identifiable Others Effect" [13] (in the following abbreviated as IOE) of which the "Identifiable Victim Effect" is a special case, and it describes the tendency of being more generous to identified victims/person as compared to unidentified or statistical victims/person.

In the context of the dictator game, Hoffman et al. [14] found that variations in the level of the recipient's anonymity caused large and significant changes in the dictator's generosity. Similarly, creating conditions of privacy and anonymity by putting the recipient in the same or a different room (or even in different cities) showed significant effects [15, 16]. Brañas-Garza [17] found that describing the recipient as poor increased the amount the dictator was willing to give away. Showing pictures of the recipient [18] or giving them a name [19] had a similar effect. Ritov & Kogut [20] investigated altruistic behavior with dictator games where the recipient was either identified i.e., personal characteristics were revealed or not identified (no information). Their results suggest, however, that the tendency to be more well-disposed towards identified than to unidentified recipients seems to be influenced by additional factors (e.g. intergroup bias).

Small & Loewenstein [21] changed the identified person from victim to perpetrator and investigated the willingness to punish uncooperative behavior in a social dilemma. Participants were more punitive toward identified perpetrators than to unidentified ones. That is, being more benevolent to identified victims and more malevolent to identified perpetrators suggests that the IOE depends also on the role of the person. In the current study, the identified/ unidentified recipient is furthermore described as a person with or without need. This additional information may tip the IOE in one or the other direction: A person in need might be perceived as a victim, whereas a person without need might be seen as a person receiving resources for no apparent reason.

Frames—different but objectively equivalent descriptions of the same problem—affect choice behavior [22]. In the context of the dictator game, frames include, for instance, buying and selling commodities or handling coins and notes (see Engel [6] for a review). In the current paradigm amounts handed over to the recipients occur with a probability, presented as a lottery and framed either as a gain or a loss (Fig 1). The framing resembles an attribute framing [23], and we assume that in a positive frame (win) higher amounts are invested as compared to a negative frame (loss).

The *size* of a framing effect seems to depend on several additional factors. For instance, in a risky gamble framing, the probability of winning/losing an amount; the magnitude of the amount itself; or specific problem domains may account for some differences observed for the effect but also design-related issues such as between-subject versus within-subject design [23–27]. Here, we include the probabilities to win the lottery as moderators [28, 29].

Furthermore, frames may have a moderating effect on need. In their study, Diederich et al. [30] induced need in terms of a minimum requirement of points a person has to meet during

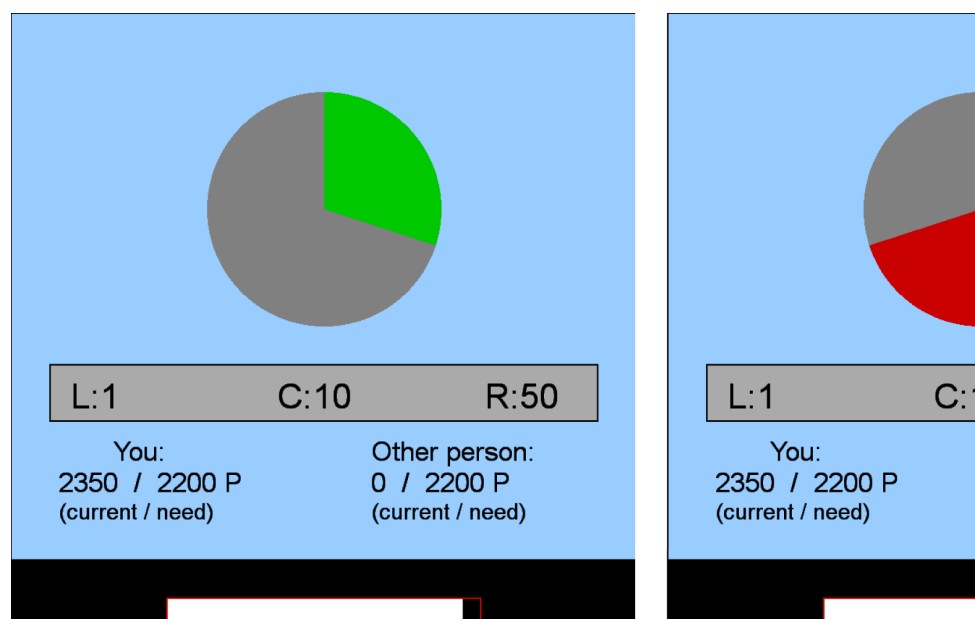

**Fig 1. Screenshots of the display.** Need game embedded in a lottery frame either as gain (left) or as loss (right). The probability of winning/losing the bet is presented in a pie chart. In a gain trial, the probability of winning is marked in green and the probability of losing in gray. In a loss trial, the probability of winning is marked in gray and the probability of losing in red. Below the pie charts are the number of points displayed from which the donor can bet on: 1, 10, or 50 points. L, C, and R in front of the bets refer to the keys the participant needs to press to indicate her/his choice. The third row in the display shows the current endowment and need of the donor ("you") and that of the recipient ("Other person"). The last row indicates the remaining time allotted to a given trial.

a series of games. Their results show that participants became more risk-taking in gain frames with increasing induced need. Mishra et al. [31] found that people who have a debt (need) that can only be settled by taking higher risks tend to show stronger framing effects than people without debt. Mishra & Fiddick [32] and Diederich et al. [29] introduced need into the Asian Disease paradigm [22]. Mishra & Fiddick defined need as a minimum number of lives the decision-maker must hypothetically save, or prevent from dying, respectively. They found that need eliminates the framing-effect. Diederich et al. investigated need, defined as the number of people who were affected by a certain disease, i.e., in need of a treatment, with various diseases (unusual disease, leukemia, AIDS). They found that when need was high, i.e., a large number of people were affected, participants became more risk-averse, and more framing-effects occurred when more people were affected by a disease. Note that the size of this moderation effect depended on the kind of disease.

Depth of processing, influenced by individual differences as well as time constraints may also affect the size of the framing effect [29, 33]. Diederich et al. and Guo et al. [29, 33] found that shorter declines enhanced the framing effect. While time constraints have shown an impact on preferential choice (frequencies) [34–36] it seems not to have a significant influence on the amount given in dictator games (for reviews see [37] and [38]). This will be probed here further.

Based on this, we hypothesize that the amount of her/his resources the donor is willing to invest in the need game depends on various factors. 1) Need: The donor is willing to invest larger amounts for a recipient in need as compared to a person without need. 2) Identifiability of the recipient: The more information about the recipient is available (i.e. identified) the higher is the amount the donor is willing to invest. 3) Need moderates the IOE: The effect is

stronger for a recipient in need. 4) Framing of the lottery: We expect that higher amounts are invested in a gain frame than in a loss frame. 5) Probabilities: The higher (lower) the probability to win (lose) the lottery is the higher are the amounts invested. 6) Time Pressure: We hypothesize that time pressure moderates the framing effect, that is, stronger framing effects under the shorter time limit. 7) We further expect framing to moderate the effects of Need (7a) and Identifiability (7b).

## Materials and methods

We start by describing the experiments' overall setup, evaluation methods and, in particular, the need game's general features.

### Need game

The need game is a modified dictator game, which includes two players, an active participant, that is, the donor, and a passive player, that is, the recipient. The donor initially owns some endowment, $K_D$, and the recipient typically has no endowment, $K_R = 0$. In the standard dictator game, the dictator decides how to split the endowment between her/himself and the recipient, who has no influence on the outcome of the game. That is, the dictator determines the portion of the endowment shared with the recipient, and the recipient receives that portion. The newly created need game differs in several ways from the standard game. First, the dictator/donor is not free of sharing any portion of the endowment with the recipient but may choose from predetermined amounts only: $x_1, \ldots, x_n$. Second, the selected amount $x_i$ is not simply handed over to the recipient but depends on the outcome of a lottery. That is, $x_i$ is a bet in a lottery with probability $p$ for the donor to win the bet and probability $1 - p$ for the donor to lose it. The rule of the game is as follows: The recipient receives the amount $x_i$ if the donor *loses* the bet. In this case, the amount lost is subtracted from the donor's endowment (i.e. $K_D - x_i$) and added to the recipient's account (i.e. $K_R + x_i$). If, however, the donor wins the bet, $x_i$ is added to the donor's endowment (i.e. $K_D + x_i$), and the recipient obtains nothing. Third, both players require a minimum need at the end of the game, $N_D$ and $N_R$, expressed in the same currency as the endowment, that they must maintain (the donor) or obtain (the recipient) by playing several rounds of the game. That is, typically at the onset of the game $K_D > N_D$ and $K_R < N_R$ and after $t$ rounds, $K_{D_t}$ must not $< N_D$ and $K_{R_t}$ should be $\geq N_R$. The recipient is paid only when his/her need threshold is obtained; likewise the donor gets paid only when his/her need threshold is maintained.

Obviously, the need game is not exactly what is defining a charity lottery. Even if they advertise their lotteries by showing individuals in need, the money goes to the charity organization as such and not necessarily to the people shown in the advertisement. Furthermore, the exact need of the people for which the organization is collecting money is not known, and typically the money is distributed among various projects rather than given to a single person (although the advertisement sometimes suggests this). Finally, whatever is collected by the charity organization is distributed, i.e. given to the needy. It does not require to meet a limit (most of the time) before the money is spent.

### Stimuli

The need game was embedded in a lottery with probabilities $p = .3, .4, .6,$ and $.7,$ of winning a bet and framed either as gain or as loss. In a gain frame, the probability $p$ of winning the lottery appeared as green area in a pie chart; the probability of losing it, $1 - p$, was coded as the remaining gray area (Fig 1, left panel). In a loss frame, the probability of winning the lottery

$p$ appeared as gray area, and the probability of losing it, $1 - p$, was marked in red (Fig 1, right panel). Note that, both lotteries are identical; only the color coding is different. Each probability was presented 25 times in each frame, amounting to a total of 200 trials.

Below the charts are shown different amounts from which the donor can select one to bet on. In the current experiment three amounts (points) were offered: $x_1 = 1$, $x_2 = 10$, and $x_3 = 50$. The third row of the display in Fig 1 shows the current endowment and the need level of the donor (left hand side and labeled as "you") and of the recipient (right hand side and labeled as "Other person"). In this example, the donor's current endowment is 2,350 points (actually, the initial endowment) and the one of the recipient is 0; the need level for both players is 2,200 points. Note that, during the experiment, the displayed endowment changes for both players after each round whereas the need display remains constant during one block of trials. At the bottom of the display is a white bar changing its size as a function of the remaining time on a given trial.

The recipient, labeled as "Other person", had three levels of identifiability. The recipients were either referred to as "Other person" only without further description or as a person described on a paper-written profile with information about name, age, place of residence, and occupation. The profile additionally contained a picture. Specifically, the person described was a 68 years old pensioner. The description is based on a real person who was also present in some conditions described in detail below. The three conditions are referred to as "no-ID", "picture-ID", and "person-ID". A description containing more details about the stimuli is provided in S1 Appendix.

## Apparatus

Stimulus presentation and response registration were controlled by one of six computer systems. The control software operated on Matlab® 2014a and 2017 including the Psychophysics Toolbox version 3.0.13 [39, 40]. The input device was a USB 2.0 button box with three buttons (L: left, C: Center, R: Right). The lotteries were represented in pie charts with a green colored proportion that indicates the winning probability in gain frames and a red colored proportion that indicates the loss probabilities in loss frames. Further information such as choice options, endowment, need, and available time for making a decision were displayed below the pie chart (see Fig 1 for screenshots of the display and S1 Appendix for more details).

## General procedure

Participants came to the Laboratory for Behavioral and Social Sciences of Jacobs University Bremen. The lab provides various cubicles and facilities for single-subject and multi-subject experiments. The participant (donor) was asked to read the instruction written on paper and report it back to the experimenter. Only after the participant had explained the task correctly in her/his own words, she/he could start the experiment. Specific information about the "other person" is given in sections Experiment 1 and Experiment 2.

The first session started with displaying information for the participants summarizing the most important instructions of the paper-written ones. After reading the information, the participants completed four guided practice trials where the outcomes of the lotteries were manipulated to demonstrate the task. After the guided practice, participants completed an additional $2 \times 4$ trials practice where they could respond freely (in the person-ID condition, the recipient was present during this time). In the first set of practice, participants had to respond within 3s and in the second set within 1s (details below). There were no time constraints during guided practice. The experimental trials started with displaying a fixation cross that lasted for 2 seconds. The subsequent screen showed the game and the time limit for that particular trial and lasted for

1 or 3 seconds, depending on the experimental condition. A response had to be made within the time limit. The last screen provided feedback about the outcome of the lottery. It was displayed for 2.5 seconds. After offset of the screen the next trial started. Missing a deadline resulted in a reduction of the donor's endowment by 10 points. The participant completed 2 sessions. One session consisted of 2 blocks, each block with 200 trials. The second session started immediately after the first session but the participant could have an optional short break between them. The experiment has been approved by the Jacobs University Research Ethics Committee.

## Statistical methods

Data were evaluated using descriptive statistics, regression analysis, and t-tests. Choice frequencies were analyzed using cumulative link mixed-effects models (CLMM) fitted with the Laplace approximation. For modeling the data, we assume an underlying probit location-scale distribution. The number of points the recipients finally obtained were evaluated using linear mixed-effects models (LMM) fitted by Restricted Maximum Likelihood (REML). Statistical analysis was performed using the computing environment R (version 3.4.2 [41], packages: 'ordinal', 'lme4', 'descr', and 'prediction'). The CLMM involved the frequencies to which the donors bet 1, 10 or 50 points in the lotteries as ordinal scaled dependent variable. As potential structure for the threshold coefficients, we assumed standard unstructured cut-points. The dependent variable of the LMM was the number of points the recipient obtained at the end of each block of trails. Explanatory variables (possible categories in parentheses) included Need (0 points; 2,200 points), Identity (recipient: no-ID; picture-ID; person-ID), Frame (loss; gain) Time-limit (1s; 3s) and Probability of winning (.3; .4; .6; .7). First categories served as reference in the statistical analysis. Furthermore, participants and experimental blocks were included as random effects to reduce the type I error rate [42] and the impact of potential sequential effects. Model 1 determines main effects for testing hypotheses H1, H2, H4, and H5. Model 2 considers 2-way interaction effects for testing hypotheses H3, H6, and H7. Finally, we performed t-tests to analyze the frequencies of blocks in which the recipient obtained at least 2,200 points (need score).

## Experiment 1

The first experiment tests the seven hypotheses using a mixed design with two identifiability levels (no-ID and picture-ID) and two levels of need (0 and 2,200) between subjects as well as Frame (Gain, Loss) and Time (1s, 3s) within subjects.

### Participants

105 students from Jacobs University Bremen and University of Bremen (52 female; age between 18 and 42 years; median = 21 years) were recruited as voluntary participants serving as donors. They gave their written informed consent prior to their inclusion in the study. Participants had the right to leave the experiment at any time without giving any reason. They were screened for their ability to follow the experimental instructions. Participants received an 8.00 € show-up fee. Furthermore, both the participants and the person providing the picture received 0.01 Euros per 15 points won in each block, provided the need level had been reached at the end of the 200 trials. Donors received on average 16.48 € (including show-up fee) and the recipient 3.31 €.

### Design and specific procedure

The Identity (no-ID; picture-ID) × Need (0 points; 2,200 points) between-subject design results in four experimental groups, and the participants were randomly assigned to these

groups: 1) no-ID recipient/0-Need (23 participants; 11 females); 2) picture-ID recipient/0-Need (26; 14); 3) no-ID recipient/2,200-Need (35; 15); 4) picture-ID recipient/2,200-Need (21; 13).

Within each group the factors Frame (Gain, Loss) × Time (1s, 3s) were presented, resulting in four experimental conditions. Within each of the four conditions, the four different probabilities of winning the lottery (.3; .4; .6; .7) were replicated 25 times and presented in random order, resulting in 100 trials per condition. One experimental block consisted of 200 trials: 100 gain trials and 100 loss trials, presented in random order. One session consisted of 2 experimental blocks: one block with time limit 1s and one block with time limit 3s, with the 3s time limit condition, presented first followed by the 1s time limit condition. Each participant performed 2 sessions, amounting to a total of 800 trials per participant.

For a given experimental group Need was the same for both players (i.e., $N_D = N_R = 2, 200$ points or $N_D = N_R = 0$ points) and the same throughout the experiment. The need level applied to one block of trials, i.e., altogether to be reached four times for each participant.

## Results

Of the 420 blocks of trials, data of 411 blocks performed by 105 participants are included in the analysis. Data of 9 blocks were lost due to a computer failure; 383 trials were timeouts resulting in 81,817 observations.

Overall the 1-point option was chosen most often (45.2%), followed by the 50-point option (32.7%) and the 10-point option was the least chosen one (22.1%). A detailed analysis is provided in S1 Appendix.

The cumulative mixed-effects model analysis included Need, Identity, Frame, Probability and Time as fixed factors and participant and experimental block as random factors.

Need and Identity had no effect on choice frequencies for specific bets, rejecting H1 and H2, respectively. Statistically significant main effects were found for Frame and Probability to win, supporting hypotheses H4 and H5. In particular, Frame had an effect on the frequency of the chosen amount with higher bets placed in gain frames than in loss frames. The effects of Probability to win also had a significant effect: the higher the probabilities to win the lotteries (the lower the probabilities to lose it), the higher were the bets placed by the participant. The results of the analysis are shown in Table 1, Model 1.

Including interactions we found significant effects for Frame × Need and Frame × Identity, supporting H7, but not for Need × Identity and Frame × Time, rejecting H3 and H6, respectively (see Table 1, Model 2).

Obviously, Need has no direct effect on choice behavior but moderates the framing effect. Albeit the framing effect is prevalent across the need conditions, it was stronger in the 0-Need condition as compared to the 2,200-Need condition. Furthermore, the interaction between Identity and Frame shows that the effect of Identifiability is more pronounced in loss frames than in gain frames. In the previous analysis, we included the amount bet on in each trial as dependent variable. It is possible that rather than trial by trial fluctuations the final amount after 200 rounds that is possibly handed over to the recipient may serve as a better indicator of the donors' willingness to invest points in lotteries in order to benefit the recipient. To test whether the amount accumulated at the end of each block is influenced by Identity and Need, we performed a linear mixed-effects model analysis accordingly. As before, Need, Identity, and Time were included as fixed factors and participants as random factor. However, the analysis showed that neither Identity nor Need nor Time had an impact on the final amount of points received by the recipients (see Table 2).

**Table 1. Experiment 1: Cumulative link mixed-effects Model 1 and Model 2.**

| Factor: | Model 1 | | | | Model 2 | | | |
|---|---|---|---|---|---|---|---|---|
| | Est. | SE | z-value | p-value | Est. | SE | z-value | p-value |
| Need (2,200) | −.069 | .102 | −.677 | .499 | .042 | .138 | .304 | .761 |
| Identity (picture-ID) | .160 | .102 | 1.569 | .117 | .304 | .151 | 2.010 | .044 |
| Frame (gain) | .486 | .009 | 53.942 | <.001 | .594 | .019 | 31.476 | <.001 |
| Probability (.4) | .362 | .013 | 27.292 | <.001 | .362 | .013 | 27.293 | <.001 |
| Probability (.6) | 1.530 | .013 | 116.552 | <.001 | 1.531 | .013 | 116.580 | <.001 |
| Probability (.7) | 2.130 | .014 | 152.709 | <.001 | 2.131 | .014 | 152.735 | <.001 |
| Time (3s) | .045 | .015 | 2.946 | <.001 | .055 | .018 | 3.115 | .002 |
| Need ×Identity | | | | | −.152 | .204 | −.745 | .456 |
| Frame×Need | | | | | −.081 | .018 | −4.457 | <.001 |
| Frame×Identity | | | | | −.118 | .018 | −6.489 | <.001 |
| Frame×Time | | | | | −.021 | .018 | −1.150 | .250 |
| Intercept 1 (1|10) | 1.124 | .094 | 12.01 | | 1.223 | .11 | 11.09 | |
| Intercept 2 (10|50) | 1.975 | .094 | 21.07 | | 2.075 | .11 | 18.79 | |

Number of observations: 81817. Groups (random effects): Participants, 105; Blocks, 4. Dependent variable: choice frequencies. Reference categories of independent variables: Need (0 points), Identity (no-ID), Frame (loss), Time (1s), and Probability (.3). Intercept 1 and 2 are threshold coefficients (cut points).

Recall that the recipients were paid only when the accumulated points during one block of 200 trials passed a preset criterion (need threshold). In the 0-Need condition, this is obtained by default and all points lost by the donor are directed to the recipient who will receive them at the end of the experiment. In the 2,200-Need condition, however, the recipient does not get anything if the threshold is not obtained.

To test whether this influences the donors' choice behavior, we inspect single blocks of trials. In 229 (182) out of the 411 blocks, participants had to maintain a need score of 2,220 points (0 points). Consider first the 2,200-Need condition. Of the 229 need-blocks, the recipient was not identified (no-ID) in 141 blocks and identified with a picture (picture-ID) in 88 blocks of trials. Of the 141 no-ID blocks, the recipient received the minimum of required 2,200 points 19 times (in about 13% of the blocks) whereas for the 88 picture-ID blocks, the minimum was obtained 27 times (in about 31% of the blocks). A t-test revealed that the proportion of recipients obtaining the required need level was higher in the picture-ID condition than in the no-ID condition ($t = 3.25$, $df = 229$, $p = .001$).

In contrast, for the 0-Need recipients in the no-ID condition (86 out of the 182 blocks) obtained the required level of 2,200 points in 13 percent and those in the picture-ID condition (the remaining 96 blocks) in 18 percent of the cases, which is statistically not significant ($t = .92$, $df = 182$, $p = .36$). The results are illustrated in Fig 2. Note that, the donors obtained 2,200 points in 95 to 97 percent of the blocks.

**Table 2. Experiment 1: Linear mixed-effects model.**

| Factor | Est. | SE | t-value | p-value |
|---|---|---|---|---|
| Need (2,200) | 6.68 | 123.016 | .054 | .957 |
| Identity (picture-ID) | 189.064 | 122.992 | 1.537 | .127 |
| Time (3s) | −12.88 | 36.306 | −.355 | .723 |
| (Intercept) | 1439.084 | 113.005 | 12.735 | < .001 |

Number of observations: 411 (blocks). Groups (random effects): Participants, 105. Dependent variable: Recipients' final block score (amount of points). Reference categories of independent variables: Need (0 points), Identity (no-ID), Time (1s).

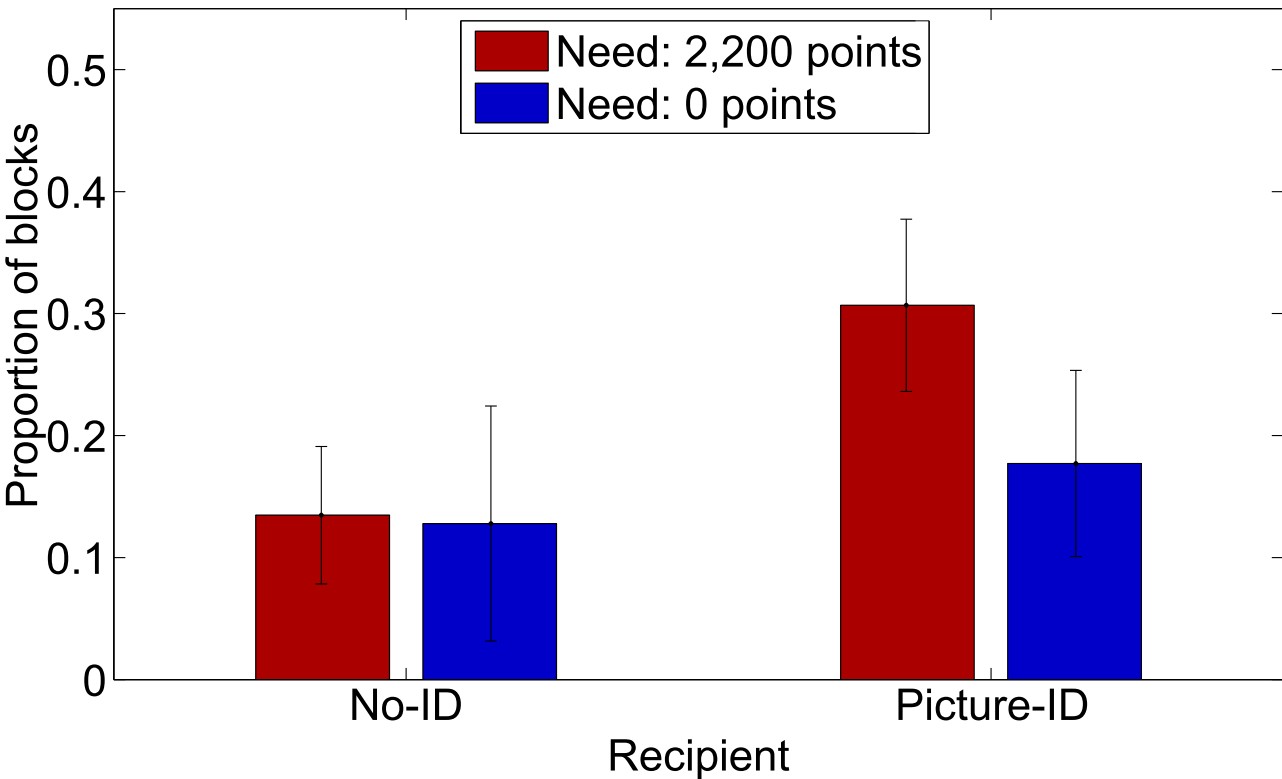

**Fig 2. Proportion (including 95% CI) of experimental blocks per Need and Idenity condition in which the recipient obtained the need level of 2,200 points.**

## Summary

Experiment 1 investigated several hypotheses about factors that may influence the donor's willingness to invest different amounts in a charity lottery-like need game. The focus was on Need and Identity. The mixed model analysis revealed no clear relationship between the bets of the donor in the lotteries and these factors. Also the final amount of points the donors send to the recipients within the experimental blocks were not influenced by either Need or Identity. However, analyzing the frequencies on how often the recipients obtained the need level showed that picture-ID recipients received a need level of 2,200 points more often than no-ID recipients but only when this was a required need level (2,200-Need condition).

Moreover, we found that donors chose to place higher bets in gain frames than in loss frames (framing effect). Not surprisingly, donors placed higher bets on lotteries when the probabilities to win were high (gain frames), or the probabilities to lose were low (loss frames). We further expected shorter response time limits to enhance the framing effect. This hypothesis was not supported by our findings. However, there was a main effect of time limits: donors placed lower bets under shorter time limits. As hypothesized, the mixed-model analysis suggests that Framing moderated the effects of identifiability and need.

## Experiment 2

To investigate the effect of identifiability in more detail, we tested a higher level of Identity to make the recipient even more identifiable to the decision-maker. For this purpose, the recipient was a real person, sitting in the same room as the donor. The within-subject conditions

were varied in the same way as in Experiment 1. Therefore, Experiment 2 can be seen as a supplementary condition to the factor Identity on the between-subject level. Since the results for the 0-Need condition in the first experiment were not particularly informative, we conducted Experiment 2 only with a need level of 2,200 points.

## Participants

Thirty-eight students from Jacobs University Bremen and University of Bremen (24 female; age between 19 and 33 years; median = 22.5 years) were recruited as voluntary participants serving as donors. The participant serving as "real person" recipient was a 68-year-old retired man recruited via a public advertisement on the bulletin board on the official website of the Free Hanseatic City of Bremen (https://schwarzesbrett.bremen.de/). He was the same person throughout the experiment. Participants had the right to leave the experiment at any time without giving any reason. Moreover, they were screened for their ability to follow the experimental instructions.

Donors completed 2 sessions which lasted for about 40 minutes. The recipient ("real person") was involved for 57 hours and was paid 8.33 Euros per hour (minimum wage) for his participation. Furthermore, both the donor and the recipient received 0.01 Euros per 15 points won in the game, provided a preset target (the need level) was maintained/obtained during 200 trials. Including show-up fee, donors received, on average, 15.65 € and the recipient 17.01 €.

## Design and specific procedure

The within-subject design with Frame (gain vs. loss), Time (3s vs. 1s), and Probability (.3, .4, .6, .7) was the same as in Experiment 1. Both participants, that is the donor and the receiver, were introduced to each other, revealing personal information about name, age, and occupation (same information as provided for the picture-ID recipient in Experiment 1). They were led into the lab-room, offered they respective seats, and given information about their role in the experiment (donor, recipient). The recipient was able to watch the donor playing the need game on a separate monitor but did not face the donor. He was instructed to be quiet during the experiments and to behave almost identically in all experimental sessions. Furthermore, he was asked to make no significant changes regarding his outward appearance between the sessions. The instruction for the donor was identical to the one in Experiment 1.

The person-ID recipient could perform up to four sessions per day. Breaks between the sessions were offered to give him time to recover (approximately 15 minutes after the first and third session and 30 minutes after the second session).

## Results

All 152 blocks of trial data performed by the 38 participants were included in the analysis. 165 trial were timeouts leaving 30,235 observations.

Different from Experiment 1, overall the 50-point option was chosen most often (40.2%), followed by the 1-point option (35.2%) and similar to Experiment 1, the 10-point option was the least chosen one (24.6%). A detailed analysis is provided in the S3 Appendix.

The mixed-effects model analysis included Frame, Probability, and Time as fixed factors (identical to the within-subject factors of Experiment 1) and participants and experimental blocks as random factors. The effects of Framing (higher bets in gain frames than in loss frames) and Probability (the higher the probabilities to win the higher the bets) were similar to those observed in Experiment 1 (see Table 3, Model 1). Also, we found no significant interaction effect of Frame × Time here either (see Table 3, Model 2).

**Table 3. Experiment 2: Cumulative link mixed-effects Model 1 and Model 2.**

| | Model 1 | | | | Model 2 | | | |
|---|---|---|---|---|---|---|---|---|
| | Est. | SE | z value | p-value | Est. | SE | z value | p-value |
| Frame (gain) | .478 | .015 | 32.569 | <.001 | .484 | .021 | 23.323 | <.001 |
| Time (3s) | −.033 | .049 | −.668 | .504 | −.026 | .051 | −.519 | .604 |
| Probability (.4) | .261 | .020 | 13.006 | <.001 | .261 | .020 | 13.006 | <.001 |
| Probability (.6) | 1.279 | .021 | 61.537 | <.001 | 1.279 | .021 | 61.537 | <.001 |
| Probability (.7) | 1.817 | .022 | 81.352 | <.001 | 1.817 | .022 | 81.351 | <.001 |
| Frame×Time | | | | | −.013 | .029 | −.431 | .667 |
| Intercept 1 (1\|10) | .508 | .131 | 3.873 | | .511 | .131 | 3.891 | |
| Intercept 2 (10\|50) | 1.443 | .131 | 10.996 | | 1.446 | .131 | 11.003 | |

Number of observations: 30,235. Groups (random effects): Participants, 38; Blocks, 4. Dependent variable: choice frequencies. Reference categories of independent variables: Frame (loss), Time (1s), and Probability (.3). Intercept 1 and 2 are threshold coefficients (cut points).

## Combined anaysis (Identity)

Because the statistical analyses showed similar results for the within-subject factors for both experiments, the different proportions of amounts bet may be caused by the identity of the recipient. That is, the physical presence of the recipient may have influenced the donor to bet on higher amounts and by that sharing resources more generously in this condition as compared to the other Identity conditions. Here we provide a combined analysis of the Identity factor. Data of 97 participants are included. Data of 3.5 sessions (7 blocks) were lost due to computer failure. 345 trials were timeouts, leaving a total of 75,855 observations (including 45,620 observations from Experiment 1 and 30,235 from Experiment 2). Table 4 shows the proportions of amounts bet on for three Identity conditions in the 2,200-Need condition. Note that, the following analysis does not involve data of the 0-Need condition. To compare choice frequencies depending on the levels of Identity, we performed a mixed-effects model analysis including Identity, Frame, Probability and Time as fixed factors and participants and experimental blocks as random factors. Table 5 shows the results with respect to Identity (complete analysis can be found in S4 Appendix). The analysis showed that donors placed significant higher bets in condition person-ID as compared to conditions no-ID and picture-ID. Effects of Framing and Probabilities were as described in Experiment 1 and 2; no significant effect of Time and no interaction effects were found (see S4 Appendix). Similar to the analysis in Experiment 1, we tested whether Identity influences the final amount of points after a block of trials (i.e. the amount possibly handed over to the recipients). A linear mixed-effects model analysis showed statistically significant effect for Identity. Person-ID recipients received higher amounts of points as compared to no-ID recipients (see Table 6).

**Table 4. Combined analysis: Relative choice frequencies per Identity condition.**

| Recipient | 1 point | 10 points | 50 points |
|---|---|---|---|
| no-ID | .47 | .23 | .31 |
| picture-ID | .46 | .19 | .35 |
| person-ID | .35 | .25 | .40 |

Overall, participants chose to bet 1 point in 42.1%, 10 points in 22.5% and 50 points in 35.4%.

**Table 5. Combined analysis: Cumulative link mixed-effects model (shortened version).**

| Factor: | Est. | SE | z-value | p-value |
|---|---|---|---|---|
| Identity (picture-ID) | .088 | .176 | .502 | .616 |
| Identity (person-ID) | .401 | .151 | 2.656 | .008 |
| Intercept 1 (1\|10) | 1.037 | .109 | 9.532 | |
| Intercept 2 (10\|50) | 1.896 | .109 | 17.413 | |

Number of observations: 75855. Groups (random effects): Participants, 97; Blocks, 4. Dependent variable: choice frequencies. Reference categories: Identity (recipient: no-ID). Intercept 1 and 2 are threshold coefficients (cut points). Full table is shown in S4 Appendix.

**Table 6. Combined analysis: Linear mixed-effects model.**

| Factor | Est. | SE | t-value | p-value |
|---|---|---|---|---|
| Identity (picture-ID) | 164.21 | 210.81 | .779 | .438 |
| Identity (person-ID) | 504.36 | 180.92 | 2.788 | .006 |
| Time (3s) | −29.01 | 41.35 | −.702 | .484 |
| (Intercept) | 1462.90 | 130.63 | 11.199 | <.001 |

Number of observations: 381 (blocks). Groups (random effects): Participants, 97. Dependent variable: Recipients' final block score (amount of points). Reference categories of independent variables: Need (0 points), Identity (no-ID), Time (1s).

As for Experiment 1, we investigated the number of blocks in which recipients obtained the need threshold of 2,200 points. The results revealed that the person-ID recipient obtained this threshold in 43 percent of the cases (66 out of 152 blocks). A pairwise t-test (Bonferroni adjusted p-values) showed that recipients obtained the required points less often in the no-ID condition as compared to the picture-ID ($t = 3.25$, $df = 229$, $p = .009$) and the person-ID condition ($t = 5.97$, $df = 293$, $p < .001$). However, no statistically significant difference was found between the relative frequencies of condition picture-ID and person-ID ($t = 1.96$, $df = 240$, $p = .14$). Fig 3 shows the respective results for the three Identity levels.

## Summary

In Experiment 2, the identified person was a real person sitting in the same room as the participant (donor). The effects of framing and different probabilities were the same as in Experiment 1: Higher bets in gain frames, and the amount of bets increased with higher probabilities to win. No effect of time limits was found.

We could show that for the person-ID recipient the donor placed higher bets as compared to the other levels of Identity (no-ID and picture-ID), which were applied in Experiment 1. A similar relationship was found between Identity and the recipients' final amount of points. At the end of a block, person-ID recipients got higher amounts than no-ID recipients.

Furthermore, we found that picture-ID and person-ID recipients obtained the required need level more often than no-ID recipients. Although the difference in proportions of obtaining the level between the picture-ID and person-ID conditions was statistically not significant, it points in the direction that the willingness to share with a real person, sitting in the same room as the donor, seems to be higher than the willingness to share with a recipient described by text and picture.

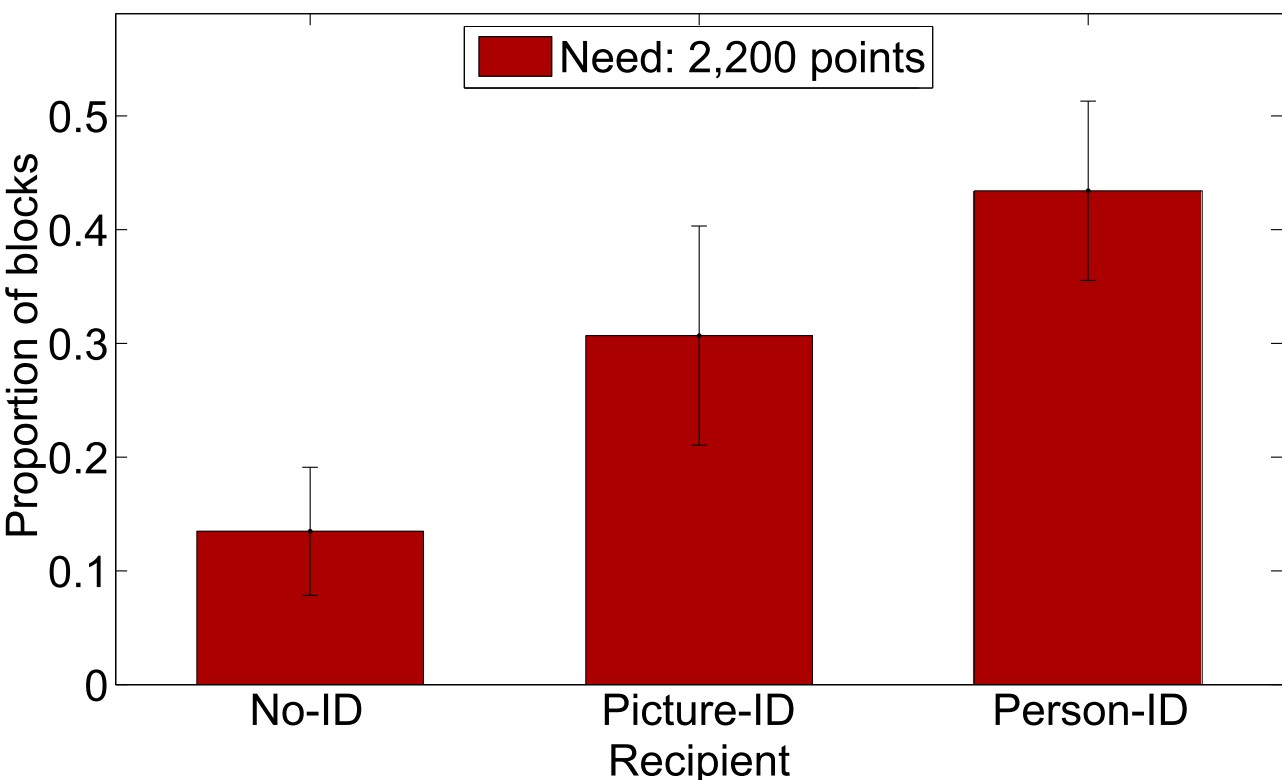

**Fig 3. Proportion (including 95% CI) of experimental blocks within each Identity level in which the recipient obtained the required need level of 2,200 points.**

## Summary and Discussion

Charity lotteries raise money for supporting social programs by selling lottery tickets to the public. While there is research on who is playing state-lotteries and why (see e.g. [43–46]) little is known whether specific properties of charity lotteries influence the player to make higher donations, i.e. to buy larger shares.

We investigated several factors that may affect players' choices in such lotteries. The motivations of gamblers in placing bets under these conditions are mixed. On the one hand, it is reasonable to assume that the gamblers strive to maximize their own gains. On the other hand, they may also care about increasing the gains for the recipient. Some of the factors we tested cannot unambiguously point to one motivation or another. Framing effects have been observed in standard (non-charity) gambles. Likewise, increased risk-taking with increased probability of winning can be motivated purely by self-interest. One factor that can unequivocally suggest the existence of other-oriented motivation is the nature of the recipient. If the gamblers strive only to maximize their own outcomes, then any characteristics of the recipient should not affect their choices. By contrast, to the extent that people think of buying charity lottery tickets as donation, the characteristics of the recipient are likely to affect their decisions. In particular, the identifiability of the recipient is expected to make a difference.

Earlier research consistently showed greater generosity towards identifiable recipients, in contexts of charitable giving. However, this effect has typically been demonstrated in situations that do not involve risk. In the present paper, we explore whether this effect extends to risky choices in charity-lotteries. Importantly, greater generosity towards identifiable others seems

to be constraint by the recipients' neediness. In a study of willingness to donate, Kogut & Ritov [10] found that identifiability increased willingness to contribute only when the recipient was needy. This increase was mediated by amplified emotional reaction to the needy victim.

In the current research, we explored the effect of recipient identifiability on choice of betting in a charity-like lottery, focusing on the need of the person to whom parts of the revenue are going. Extrapolating from research on donation, we conjectured that identifying the recipient would affect betting choices mostly when the recipient is in need.

We conducted two experiments and proposed a new experimental paradigm that we called "need game". It involves two players—a donor and a recipient—for investigating the effects of identifiability and need, as well as the effects of presenting the lotteries in a gain or loss frame, the effects of different probabilities to win/lose the lottery, and time limits for the response. The donor was the decision-maker endowed with some initial amount of points (to be converted to money), and the recipient was a passive participant with no endowment who was not able to intervene in the game (similar to a dictator game). The need game is played for several rounds in which a preset need threshold (points, converted to money) is to be maintained (donor) and obtained (recipient). Unlike the dictator game, the need game involves a lottery and amounts are transferred to the recipinet only in case the donor loses the lottery. Otherwise, the amount bet on is added to the donor's account. The focus of the current experiments was on the identifiability and need (threshold) of the recipient. In Experiment 1, we included the factor Identity with two levels to test the Identifiable Others Effect (IOE): a person described as "other person" (no-ID); and a person described with respect to some demographic information and shown in a picture (picture-ID). The factor Need had two levels: 0-points and 2,200-points (to be converted to money). Although mixed-effects model analyses showed neither a main effect for Identity nor for Need on the amounts bet in each round and thereby rejecting our hypotheses, a more detailed analysis for Experiment 1 showed an IOE (no-ID versus picture-ID) when considering how often the need threshold was obtained for the recipient. The effect was stronger when the need was high (2,220 points). In Experiment 2, we tested a higher level of Identity, in order to make the recipient even more identifiable: a real person introducing himself and present in the same room as the donor (person-ID). Need was set to a fixed value of 2,200 points. To compare the findings provided by the two experiments, we performed a combined analysis which suggests that the identifiability of the recipient played a major role in the decision on how much to bet on charity lottery-like games. This manifested itself in three ways: First, as compared to the no-ID condition, participants placed higher bets on lotteries in the person-ID condition; second, person-ID recipients finally received higher amounts of points at the end of 200 rounds (that is one experimental block) than no-ID recipients; and third, the more identifiable the recipient was (from no-ID to picture-ID to person-ID) the more often the need threshold at the end of a block was reached. The results support the findings of Jenni & Loewenstein [8] and Kogut & Ritov [9, 10] and provide a clear indication that participants' decisions were motivated not only by the goal of maximizing their own profit but also by consideration of the recipient's welfare.

To compare our results with other studies that focused on need [29, 30] we included Frame (gain, loss) and Time (1s, 3s) as additional factors. Probability to win a lottery was included to complete the design.

Results of both experiments showed strong framing effects: donors invested more points in gain than in loss lotteries which corroborates the vast amount of evidence in framing research (see e.g. [23, 25, 27]). That is, when the lottery is framed as a gain (win) then the donor bets higher amounts, possibly assuming to win the bet. When the lottery is framed as a loss, smaller amounts are bet on, possibly assuming to lose the bet (see e.g. [24]). Note that the framing applied here does not strictly belong to any of the categories as proposed by Levin et al. [23].

Although we speak of gain and loss lotteries here, a risky-choice framing typically includes a choice between a risky and a sure option (e.g. Asian Disease Problem [22]). In those situations, the decision-maker typically tends to prefer the risky option in a loss frame and the sure option in a gain frame. In the current setting, there is no choice between two lottery options but between different amounts to bet with probabilities framed as gains or loss. It resembles more an attribute framing but strictly speaking adds another variation of framings.

Previous research has shown interactions between frames and available time for making a choice in risky-choice tasks [29, 30, 33] with shorter time limits enhancing the framing effect.

In the present study, we found a main effect for Time in Experiment 1: participants placed lower bets under the shorter time limit (1s) but no significant interaction between different time limits and framings.

Not surprisingly, we observed that donors placed higher bets on lotteries when the probabilities to win were high (gain frames), or the probabilities to lose were low (loss frames).

We further expected that Need and Identity serve as moderators for framing effects. Only the results of Experiment 1 support the hypotheses: We observed slightly stronger framing effects among the 0-Need condition. This result supports the findings of Diederich et al. [30] and Mishra and Fiddick [32]. Furthermore, stronger framing effects were found among participants who were assigned to the no-ID condition of Identity. However, this was not supported by the combined analysis considering only participants who were assigned to the 2,200-Need condition. A possible explanation for this could be provided by a three-way interaction effect involving the factors Frame, Need, and Identity. Such a relationship, however, was neither hypothesized nor tested here.

To summarize: Which factors influence the willingness to invest resources in charity lotteries? Overall, we found that the amount of sharing was increased for identified recipients. This finding provides a clear indication that participants' decisions were motivated not only by the goal of maximizing their own profit but also by consideration of the recipient's welfare. There was no main effect of Need but we found that a preset need level gets donors to satisfy the need of identified recipients more often. Unidentified recipients did not benefit from a need level. This could indicate that the status of neediness is more likely to be perceived, recognized or considered in the decision to donate when the person in need is not anonymous. Moreover, strong framing effects were observed: donors invested more points in gain than in loss lotteries. The investment increases (decreases) with the probability to win (lose). Contrary to our expectations, time limits did not moderate framing effects in the charity lottery-like need game.

Final remarks: The newly created need game may open up for investigating the willingness to share resources with needy people. For instance, it may depend on how similar donors and recipients are, how much they would like each other, and so on, basically determinants known from social psychology on group identity [47, 48]. Further framings applied in the context of the dictator game may be incorporated such as wordings like "boost" or "donate" [49] or the currency used [50]. Finally, individual differences in personality among the donors may be included as explanatory variables, (e.g. [51]) to account for differences in the willingness to share.

## Supporting information

**S1 Appendix. Stimuli and apparatus in detail.**
(PDF)

**S2 Appendix. Descriptives of experiment 1 in detail.**
(PDF)

**S3 Appendix. Descriptives of experiment 2 in detail.**
(PDF)

**S4 Appendix. Combined analysis (mixed-effects models) of experiment 1 and 2 in detail.**
(PDF)

## Author Contributions

**Conceptualization:** Adele Diederich.

**Data curation:** Marc Wyszynski.

**Formal analysis:** Marc Wyszynski, Adele Diederich.

**Funding acquisition:** Adele Diederich.

**Investigation:** Marc Wyszynski.

**Methodology:** Marc Wyszynski.

**Project administration:** Adele Diederich.

**Resources:** Adele Diederich.

**Software:** Marc Wyszynski.

**Supervision:** Adele Diederich.

**Validation:** Adele Diederich, Ilana Ritov.

**Visualization:** Marc Wyszynski.

**Writing – original draft:** Marc Wyszynski, Adele Diederich, Ilana Ritov.

**Writing – review & editing:** Marc Wyszynski, Adele Diederich, Ilana Ritov.

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
