## [Decision Letter · Decision Letter 0]

24 Apr 2020

PONE-D-20-08877

Gamble for the needy! Does identifiability enhance generosity?

PLOS ONE

Dear Mr. Wyszynski,

Thank you for submitting your manuscript to PLOS ONE. After careful consideration, we feel that it has merit but does not fully meet PLOS ONE’s publication criteria as it currently stands. Therefore, we invite you to submit a revised version of the manuscript that addresses the points raised during the review process.

We would appreciate receiving your revised manuscript by Jun 08 2020 11:59PM. To enhance the reproducibility of your results, we recommend that if applicable you deposit your laboratory protocols in protocols.io, where a protocol can be assigned its own identifier (DOI) such that it can be cited independently in the future. For instructions see: http://journals.plos.org/plosone/s/submission-guidelines#loc-laboratory-protocols

We look forward to receiving your revised manuscript.

Kind regards,

Valerio Capraro

Academic Editor

PLOS ONE

Additional Editor Comments (if provided):

I have now collected one review from one expert in the field. I was unable to find a second reviewer, but the one I could find provided a very informative review. Moreover, I am myself very familiar with the topic, having done directly related research, therefore I feel comfortable making a decision with only one review plus my own judgment. The reviewer is positive but suggests several changes. I agree with them. Therefore, I would like to invite you to revise your work following their comments. Moreover, I would like to add the following comments:

1) Discussion about framing effects in the DG. I've done myself quite a lot of work on this topic recently, showing that moral frames dramatically impact DG behaviour (Capraro & Vanzo, 2019; Capraro et al. 2019; Bilancini et al. 2020). Of course, citing these works is not a requirement, but I'm mentioning them because they seem very related to your discussion.

2) Discussion about the effect of time pressure on DG behaviour. Sorry to say this, but this part is largely incomplete. I happen to know this research very well (see my review: Capraro, 2019). The paper Cappelen et al. is a correlational paper and not a causal paper. The review by Evans et al also look at the correlations plus it is about cooperation in the public goods game and not altruism in the dictator game. Your setting is different: time pressure and DG. You correctly acknowledge that imposing a time limit and looking at response times are different thing, but you did not mention at all the huge literature on time pressure and DG. In this setting, there are two meta-analyses (Rand et al. 2016; Fromell et al 2020) and a review (Capraro, 2019), plus many individual papers.

I am looking forward for the revision.

References

Bilancini, E., Boncinelli, L., Capraro, V., Celadin, T., & Di Paolo, R. (2020). " Do the right thing" for whom? An experiment on ingroup favouritism, group assorting and moral suasion. Judgment and Decision Making, 15, 182-192.

Capraro, V., & Vanzo, A. (2019). The power of moral words: Loaded language generates framing effects in the extreme dictator game. Judgment and Decision Making, 14, 309-317.

Capraro, V., Jagfeld, G., Klein, R., Mul, M., & van de Pol, I. (2019). Increasing altruistic and cooperative behaviour with simple moral nudges. Scientific reports, 9, 11880.

Capraro, V. (2019). The dual-process approach to human sociality: A review. Available at SSRN 3409146.

Fromell, H., Nosenzo, D., & Owens, T. (2020). Altruism, Fast and Slow? Evidence from a meta-analysis and a new experiment. Experimental Economics, 1-23.

Rand, D. G., Brescoll, V. L., Everett, J. A., Capraro, V., & Barcelo, H. (2016). Social heuristics and social roles: Intuition favors altruism for women but not for men. Journal of Experimental Psychology: General, 145(4), 389.

Reviewers' comments:

Reviewer's Responses to Questions

**Comments to the Author**

1. Is the manuscript technically sound, and do the data support the conclusions?

Reviewer #1: Yes

2. Has the statistical analysis been performed appropriately and rigorously? 

Reviewer #1: Yes

3. Have the authors made all data underlying the findings in their manuscript fully available?

Reviewer #1: Yes

4. Is the manuscript presented in an intelligible fashion and written in standard English?

Reviewer #1: Yes

5. Review Comments to the Author

Reviewer #1: Overall, I thought this was an interesting article with a new game that has the potential to study new and interesting behavioral questions. The methods and analyses are strong. I believe this will be a valuable addition to the literature, once some concerns are addressed.

Major points

- The payment structure of the game is unclear. The way I am understanding it is that if the donor meets their minimum need they are paid, regardless if the recipient’s need is met? Likewise, the recipient is paid if their need is met, irrespective of if the need is met of the donor? Or are they both paid if the recipient’s minimum need is met? If it is the latter, the participants in the game are in an interdependent interaction, where the outcomes of both players are dependent on the choices made by the decider. However, charity-lotteries are not interdependent interactions – the outcome of the charity is not dependent on the outcome of the individual who chooses to participate in the lottery. The degree of interdependence has important consequences in how and why people make decisions, which would be different for charity-lotteries and two-player dynamics.

- The methods are unclear. There are too many details in the main text that are not necessary to understand the study, such as the apparatus and much of the text under stimuli. Simplify to make the main text as clear as possible, and relegate all additional details to a supplement. This would greatly improve the readability of the manuscript.

- I am concerned about the framing of this game as similar to charity-lotteries. To begin, participants did not have a chance to not gamble and walk away, like they do in charity-lotteries. This constrains the choices of when and why people will choose to risk. Additionally, the recipient was an individual, not a charity. Moreover, the need of the recipient is largely unknown to people participating in charity lotteries, and they are not the sole person responsible for the outcome of the recipient. Lastly, the lottery would still receive the money spent, even if their ‘need’ is not met. These points should be discussed to clarify how the comparison between charity-lotteries and the need game can be made.

- This paper does not incorporate much theory or prior work on relevant topics, such as the literature on needs-based helping and risk-taking, which address when and why people will take risks. I’ve offered some references below to help with this. To be clear, I do not require the authors to cite these specific papers, rather I am trying to point them in relevant directions to strengthen the paper. Authors could include the suggested papers if they feel it would strengthen their arguments.

- The paper would benefit from a paragraph or two describing how this game could address new and interesting research questions beyond the ones presented in the current paper.

Minor points

- The title of the article (or abstract) is misleading. What is the main focus of the paper? The title suggests that identifiability is the main focus, but identifiability is not even mentioned in the abstract. Likewise, the abstract is solely focused on the game, and none of the tested predictions in the paper. These both need to be re-framed to focus on the main points of the paper.

- The fact that if the minimum need is not met both players receive nothing needs to be noted earlier, including in the abstract. This point was unclear, and should be noted earlier to really understand the game.

- What do you mean by ‘relative deprivation’ on p.2, line 35? This term is used differently by different groups of researchers.

References

Needs-based helping

- Cronk, L., Berbesque, C., Conte, T., Gervais, M., Iyer, P., McCarthy, B., ... & Aktipis, A. (2019). Managing risk through cooperation: Need-based transfers and risk pooling among the societies of the Human Generosity Project. In Global Perspectives on Long Term Community Resource Management (pp. 41-75). Springer, Cham.

- Hao, Y., Armbruster, D., Cronk, L., & Aktipis, C. A. (2015). Need-based transfers on a network: a model of risk-pooling in ecologically volatile environments. Evolution and Human Behavior, 36(4), 265-273.

- Aktipis, A., De Aguiar, R., Flaherty, A., Iyer, P., Sonkoi, D., & Cronk, L. (2016). Cooperation in an uncertain world: For the Maasai of East Africa, need-based transfers outperform account-keeping in volatile environments. Human Ecology, 44(3), 353-364.

Risk-taking

- Mishra, S., Barclay, P., & Sparks, A. (2017). The relative state model: Integrating need-based and ability-based pathways to risk-taking. Personality and Social Psychology Review, 21(2), 176-198.

- Barclay, P., Mishra, S., & Sparks, A. M. (2018). State-dependent risk-taking. Proceedings of the Royal Society B: Biological Sciences, 285(1881), 20180180.

Interdependence

- Balliet, Daniel, Joshua M. Tybur, and Paul AM Van Lange. "Functional interdependence theory: An evolutionary account of social situations." Personality and Social Psychology Review 21.4 (2017): 361-388.

- Aktipis, A., Cronk, L., Alcock, J., Ayers, J. D., Baciu, C., Balliet, D., ... & Sullivan, D. (2018). Understanding cooperation through fitness interdependence. Nature Human Behaviour, 2(7), 429-431.

Framing Effects in Ultimatum Game

- Lightner, A. D., Barclay, P., & Hagen, E. H. (2017). Radical framing effects in the ultimatum game: the impact of explicit culturally transmitted frames on economic decision-making. Royal Society open science, 4(12), 170543.

6. PLOS authors have the option to publish the peer review history of their article (what does this mean?). If published, this will include your full peer review and any attached files.

Reviewer #1: No

---

## [Author Response · Author response to Decision Letter 0]

5 May 2020

Dear Dr. Capraro,

We are very grateful for an extremely constructive and thoughtful set of reviews and the opportunity to resubmit. We append a detailed record of all the changes below. 

Thank you 

Marc Wyszynski on behalf of all authors 

Response to Editor and Reviewer

(Replies start with *)

Editor comments:

1) Discussion about framing effects in the DG. I've done myself quite a lot of work on this topic recently, showing that moral frames dramatically impact DG behaviour (Capraro & Vanzo, 2019; Capraro et al. 2019; Bilancini et al. 2020). Of course, citing these works is not a requirement, but I'm mentioning them because they seem very related to your discussion.

*Thank you very much for referring to your interesting work. We took it up in the discussion pointing to future directions. 

2) Discussion about the effect of time pressure on DG behaviour. Sorry to say this, but this part is largely incomplete. I happen to know this research very well (see my review: Capraro, 2019). The paper Cappelen et al. is a correlational paper and not a causal paper. The review by Evans et al also look at the correlations plus it is about cooperation in the public goods game and not altruism in the dictator game. Your setting is different: time pressure and DG. You correctly acknowledge that imposing a time limit and looking at response times are different thing, but you did not mention at all the huge literature on time pressure and DG. In this setting, there are two meta-analyses (Rand et al. 2016; Fromell et al 2020) and a review (Capraro, 2019), plus many individual papers.

*We agree that our setting is different. With respect to time, time pressure, and dual processes we would not agree with Rand et al. (see also the criticism by Ian Krajbich and an article on modeling time and dual processes (Diederich & Trueblood, Psych Rev, 2018, Vol. 125, No. 2, 270–292)). For not confusing the reader we dropped the entire part on reaction time. Nevertheless, your review article appears relevant.

Reviewer #1:

Overall, I thought this was an interesting article with a new game that has the potential to study new and interesting behavioral questions. The methods and analyses are strong. I believe this will be a valuable addition to the literature, once some concerns are addressed.

*Thank you very much for the kind remarks. 

The payment structure of the game is unclear. The way I am understanding it is that if the donor meets their minimum need they are paid, regardless if the recipient’s need is met? Likewise, the recipient is paid if their need is met, irrespective of if the need is met of the donor?

*This is correct. We are sorry for any confusion. We explicitly state this now in the description of the rules. 

The methods are unclear. There are too many details in the main text that are not necessary to understand the study, such as the apparatus and much of the text under stimuli. Simplify to make the main text as clear as possible, and relegate all additional details to a supplement. This would greatly improve the readability of the manuscript.

*Thank you very much for the suggestions. We moved the detailed descriptions to the supplement.

---

## [Decision Letter · Decision Letter 1]

18 May 2020

PONE-D-20-08877R1

Gamble for the needy! Does identifiability enhances donation?

PLOS ONE

Dear Mr. Wyszynski,

Thank you for submitting your manuscript to PLOS ONE. After careful consideration, we feel that it has merit but does not fully meet PLOS ONE’s publication criteria as it currently stands. Therefore, we invite you to submit a revised version of the manuscript that addresses the points raised during the review process.

We would appreciate receiving your revised manuscript by Jul 02 2020 11:59PM. To enhance the reproducibility of your results, we recommend that if applicable you deposit your laboratory protocols in protocols.io, where a protocol can be assigned its own identifier (DOI) such that it can be cited independently in the future. For instructions see: http://journals.plos.org/plosone/s/submission-guidelines#loc-laboratory-protocols

We look forward to receiving your revised manuscript.

Kind regards,

Valerio Capraro

Academic Editor

PLOS ONE

Additional Editor Comments (if provided):

The reviewer is pleased by the improvements made to the paper, but suggests a few minor changes to further improve the manuscript before publication. Please address these last comments at your earliest convenience. I am looking forward to the final version.

Reviewers' comments:

Reviewer's Responses to Questions

**Comments to the Author**

1. If the authors have adequately addressed your comments raised in a previous round of review and you feel that this manuscript is now acceptable for publication, you may indicate that here to bypass the “Comments to the Author” section, enter your conflict of interest statement in the “Confidential to Editor” section, and submit your "Accept" recommendation.

Reviewer #1: (No Response)

2. Is the manuscript technically sound, and do the data support the conclusions?

Reviewer #1: Yes

3. Has the statistical analysis been performed appropriately and rigorously? 

Reviewer #1: Yes

4. Have the authors made all data underlying the findings in their manuscript fully available?

Reviewer #1: Yes

5. Is the manuscript presented in an intelligible fashion and written in standard English?

Reviewer #1: Yes

6. Review Comments to the Author

Reviewer #1: Article: Gamble for the needy! Does identifiability enhance generosity?

Plos one

I am pleased to see the vast improvements in the manuscript, which has resulted in a much clearer paper. I still have a few minor comments, which I hope help improve the paper further.

1. When you create parallels between the need-game and charity lotteries, the connection needs to be made clear for the reader. Why is this game appropriate for studying charity-lotteries? As I pointed out in my last review, there are many differences, so highlight the similarities when you introduce these ideas (abstract, intro), otherwise it is difficult to draw the connection. In particular, the paragraph starting at line 23, p.2 starts with charities lotteries, then discusses a specific recipient. There seems to be a disconnect.

2. The description of the interaction effects in the results section is not sufficient to fully understand what is going on. For example, do you mean that for the 2200-need condition in experiment 1, people still made higher bets in gain frames than on the 0-need condition? Perhaps a figure would make this easier to understand.

3. I would recommend putting all appendices into a single document for ease of access and reading.

7. PLOS authors have the option to publish the peer review history of their article (what does this mean?). If published, this will include your full peer review and any attached files.

Reviewer #1: No

---

## [Author Response · Author response to Decision Letter 1]

22 May 2020

Dear Dr. Capraro,

Thank you again for handling our article and for the opportunity to make further changes that will surely improve the article. We append a detailed record of all the changes below. 

Sincerely,

Marc Wyszynski on behalf of all authors 

Response to Reviewer

(Replies start with *)

Reviewer #1:

I am pleased to see the vast improvements in the manuscript, which has resulted in a much clearer paper. I still have a few minor comments, which I hope help improve the paper further.

*Thanks again for your kind and constructive remarks. 

1. When you create parallels between the need-game and charity lotteries, the connection needs to be made clear for the reader. Why is this game appropriate for studying charity-lotteries? As I pointed out in my last review, there are many differences, so highlight the similarities when you introduce these ideas (abstract, intro), otherwise it is difficult to draw the connection. In particular, the paragraph starting at line 23, p.2 starts with charities lotteries, then discusses a specific recipient. There seems to be a disconnect.

*Thank you for this suggestion. We modified the respective paragraph to make the connection clearer.

2. The description of the interaction effects in the results section is not sufficient to fully understand what is going on. For example, do you mean that for the 2200-need condition in experiment 1, people still made higher bets in gain frames than on the 0-need condition? Perhaps a figure would make this easier to understand.

*Yes, the framing effect is observable across both need conditions. It was stronger without a need. We added a further note which should better describe this effect.

3. I would recommend putting all appendices into a single document for ease of access and reading.

*Thank you for the recommendation. After careful consideration, we decided not to combine the appendices because separate documents make the reference to the respective appendix clearer when reading the body text (e.g. through the hyperref-function, which is activated in the PlosOne-Latex template). Furthermore, the content of the appendices is clearly listed in the "supporting information" section. We believe, therefore, that a single document would not improve accessibility and clearness.

---

## [Editor Report · Decision Letter 2]

26 May 2020

Gamble for the needy! Does identifiability enhances donation?

PONE-D-20-08877R2

Dear Dr. Wyszynski,

We are pleased to inform you that your manuscript has been judged scientifically suitable for publication and will be formally accepted for publication once it complies with all outstanding technical requirements.

With kind regards,

Valerio Capraro

Academic Editor

PLOS ONE
---

## [Editor Report · Acceptance letter]

4 Jun 2020

PONE-D-20-08877R2 

Gamble for the needy! Does identifiability enhances donation? 

Dear Dr. Wyszynski:

I'm pleased to inform you that your manuscript has been deemed suitable for publication in PLOS ONE. Congratulations! Your manuscript is now with our production department. 

Kind regards, 

on behalf of

Dr. Valerio Capraro 

Academic Editor

PLOS ONE